# Association of Circulating Irisin Concentrations with Weight Loss after Roux-en-Y Gastric Bypass Surgery

**DOI:** 10.3390/ijerph16040660

**Published:** 2019-02-24

**Authors:** Yeon Ji Lee, Yoonseok Heo, Ji-Ho Choi, Sunghyouk Park, Kyoung Kon Kim, Dong Wun Shin, Ju-Hee Kang

**Affiliations:** 1Department of Family Medicine, Obesity Center, Inha University Hospital, Inha University, Incheon 22332, Korea; dawndusk@inha.ac.kr (Y.J.L.); wisdom@inha.ac.kr (J.-H.C.); 2Department of Surgery, Inha University Hospital, Inha University, Incheon 22332, Korea; gshur@inha.ac.kr; 3College of Pharmacy, Natural Product Research Institute, Seoul National University, Seoul 08826, Korea; psh@snu.ac.kr; 4Department of Family Medicine, Gil Hospital, Gachon School of Medicine, Incheon 21565, Korea; zaduplum@gilhospital.com; 5Department of Emergency Medicine, Inje University Ilsan Paik Hospital, Gyeonggi-Do 10380, Korea; 6Department of Pharmacology and Hypoxia-related Disease Research Center, College of Medicine, Inha University, Incheon 22212, Korea

**Keywords:** bariatric surgery, irisin, myokine, obesity, Roux-en-Y gastric bypass, weight loss

## Abstract

Irisin is a myokine with potential anti-obesity properties that has been suggested to increase energy expenditure in obese patients. However, there is limited clinical information on the biology of irisin in humans, especially in morbidly obese patients undergoing bariatric surgery. We aimed to assess the association of circulating irisin concentrations with weight loss in obese patients undergoing bariatric surgery. This was a pilot, single-centre, longitudinal observational study. We recruited 25 morbidly obese subjects who underwent Roux-en-Y gastric bypass surgery (RYGBP), and blood samples from 12 patients were taken to measure serum irisin concentrations before, and one and nine months after surgery. Their clinical characteristics were measured for one year. The preoperative serum irisin concentration (mean 1.01 ± 0.23 μg/mL, range 0.73–1.49) changed bidirectionally one month after RYGBP. The mean concentration at nine months was 1.11 ± 0.15 μg/mL (range 0.92–1.35). Eight patients had elevated irisin levels compared with their preoperative values, but four did not. Elevations of irisin levels nine months, but not one month, after surgery, were associated with lower preoperative levels (*p* = 0.016) and worse weight reduction rates (*p* = 0.006 for the percentage excess weight loss and *p* = 0.032 for changes in body mass index). The preoperative serum irisin concentrations were significantly correlated with the percentage of excess weight loss for one year (R^2^ = 0.612; *p* = 0.04) in our study. Our results suggest that preoperative circulating irisin concentrations may be at least in part associated with a weight loss effect of bariatric surgery in morbidly obese patients. Further large-scale clinical studies are needed to ratify these findings.

## 1. Introduction

Obesity is one of the most concerning health problems in the world [1]. In addition to causing type 2 diabetes mellitus, it triggers certain forms of cancer, respiratory complications, and cardiovascular diseases, which lead to high rates of mortality and morbidity [2]. Obesity is a state of excess adiposity coupled with adiposopathy, which provokes chronic inflammation and dysregulation of energy homeostasis [3].

Brown adipose tissue (BAT) and the browning process—white-to-brown adipocyte transdifferentiation—are present in adult humans [4,5,6,7,8]. Adipocytes were formerly classified into two types: fat-storing white adipocytes and thermogenic brown adipocytes. White adipocytes store triglycerides and fatty acids with a very few mitochondria and secrete a variety of adipokines such as leptin [9,10]. In contrast, brown adipocytes are active energy consumers containing many small lipid droplets and abundant mitochondria [11,12]. BAT has been recognized as a potential target for treating obesity and metabolic diseases by promoting energy expenditure. Prolonged thermogenic stimulation, exercise, chronic adrenergic stimulation, cardiac peptides, adipokines, myokines, and hepatokines have been identified as factors that can induce the browning process [13].

Irisin is a small peptide that is cleaved from the fibronectin type III domain 5 (FNDC5), a type I transmembrane protein predominantly existing in skeletal and cardiac muscles, and is secreted into the circulation [14]. As an exercise-regulated myokine, irisin is thought to play a role in exercise-induced browning of white adipose tissue and in promoting mitochondrial biogenesis or metabolic gene expression in skeletal muscle [14,15,16,17,18,19]. Although irisin has attracted more attention than other factors because of its therapeutic potential as a direct signal to intramuscular fat, the role of circulating irisin is equivocal, as it varies with different physiological or experimental conditions, particularly in humans [20,21,22,23,24].

Bariatric surgery has shown effective body-weight reduction in morbidly obese patients along with reduced mortality [25]. The effects of bariatric surgery are supposed to be associated with humoral factors and molecular changes in multiple organs including the gastrointestinal tract and adipose tissues, as well as improvements in body composition, energy expenditure, and even neuronal signals controlling appetitive behaviour [26,27,28].

The aim of this work was to advance the clinical understanding of the role of irisin in patients undergoing bariatric surgery through a prospective pilot study and to generate hypotheses for the association between the circulating levels of irisin and weight reduction and/or the metabolic effects of such surgery.

## 2. Materials and Methods

### 2.1. Study Design and Subjects

The eligible population of this prospective cohort study was obese patients who agreed to participate in the research and who underwent standard laparoscopic Roux-en-Y gastric bypass surgery (RYGBP) [29] from July 2011 to September 2013 at the Obesity Center, Inha University Hospital in Incheon, South Korea. All agreed to surgical treatment for obesity and were informed about the related morbidity. Informed consent was obtained from all individual participants included in the study. Among the 25 patients, we had to exclude some because of missing routine follow-ups (*n* = 4), lack of research samples taken at the determined time (*n* = 7), and spontaneous pregnancy after the RYGBP (*n* = 2). Twelve patients were included in the final study. All procedures performed during this study were in accordance with the ethical standards of the national research committee and with the 1964 Helsinki declaration and its later amendments. The Institutional Review Board of Inha University Hospital approved the study (2009-1473).

### 2.2. Clinical Assessment and Collection of Research Samples

A trained nurse at the obesity center measured the anthropometric data. Computed tomography (CT) scans and dual-energy X-ray absorptiometry (DEXA) were performed twice, before and one year after RYGBP. Before surgery, blood samples were obtained from the patients following a 12 h fast for the measurement of metabolic parameters. Patients visited the obesity center 2 weeks, 1, 3, 6, 9, and 12 months after RYGBP. Anthropometric and metabolic parameters were measured every three months. The blood samples were centrifuged at room temperature at 2000 xg for 10 min immediately after sampling, and the plasma samples were stored in microcentrifuge tubes at less than −20 °C until analysis.

### 2.3. Assessment of Circulating Irisin

Serum concentrations of irisin (a fragment of FNDC5) were measured using commercial enzyme-linked immunosorbent assay kits (RAG018R, BioVendor Inc., Candler, NC, USA) according to the manufacturer’s instructions. The lowest level of irisin that can be detected by this assay is 1 ng/mL, and the assay range is 0.001–5 μg/mL. The intra- and inter-assay coefficients of variance are 5–8% and 8–10%, respectively.

### 2.4. Statistical Analysis

Data are presented as the mean ± standard error, and nonparametric methods were applied. The Mann–Whitney *U* test was used for comparing mean values between groups. The Wilcoxon matched pairs signed-rank test, Friedman’s F test, or two-way repeated measures analysis of variance (RM ANOVA) were used to compare anthropometric variables, metabolic parameters, and serum irisin concentrations among the repeated measures, respectively. Spearman’s rank correlation test and linear regression analysis were employed to assess the associations between irisin levels and anthropometric or metabolic parameters. We used SPSS (version 19.0 for Windows; SPSS Inc., Chicago, IL, USA) for all statistical analyses, and *p* < 0.05 was regarded as statistically significant.

## 3. Results

### 3.1. Longitudinal Changes in Clinical Characteristics

The preoperative characteristics of the 12 study subjects, including their anthropometric variables and laboratory data, are listed in Table 1. All were morbidly obese with a mean body mass index (BMI) of 40.6 ± 4.2 kg/m^2^ (range 30.2–45.9) and a mean age of 37.6 ± 10.7 years (range 25–56). Among them, five were men, three had diabetes mellitus, and eight had taken antihypertensive medication.

One year after RYGBP, the mean weight loss was 33.8 ± 10.5 kg, with a 29.5 ± 8.1% loss of body weight, and a 79.4 ± 19.6% excess weight loss (%EWL). The weight loss during the first three months was 21.5 ± 5.4 kg (67.7 ± 19.8% of total body weight reduction) with 52.7 ± 17.9 %EWL. For whole-body DEXA (dual-energy x-ray absorptiometry) when analysing body composition, the preoperative total fat percentage was 48.1 ± 7.5%, with 55.0 ± 11.4 kg of total fat mass (TFM), and this reduced to 33.6 ± 8.1% with 27.0 ± 8.3 kg of TFM one year postoperatively. Lean body mass (LBM) decreased from 57.5 ± 12.4 kg to 51.3 ± 12.2 kg one year postoperatively. In CT measurements for abdominal fat distribution, the preoperative total fat area (TFA) of 691.7 ± 220.5 cm^2^ and visceral fat area (VFA) of 183.5 ± 80.4 cm^2^ decreased to 311.1 ± 133.6 cm^2^ and 45.7 ± 29.5 cm^2^, respectively. Along with losing body weight and decreasing adiposity, the metabolic parameters improved significantly. None of the three diabetic patients needed anti-diabetic medication or insulin to control blood glucose levels from one week to one year after RYGBP.

### 3.2. Longitudinal Changes in Circulating Irisin Levels

The preoperative circulating serum irisin concentration was 1.01 ± 0.23 μg/mL, with a range of 0.73–1.49 μg/mL. The concentrations altered bidirectionally in this group one month after RYGBP, within a range of 0.55–2.51 μg/mL. This dispersal disappeared nine months after RYGBP, with a mean value of 1.11 ± 0.15 μg/mL and a range of 0.92–1.35 μg/mL (Figure 1).

We observed two groups displaying either increases or decreases in irisin levels one or nine months after surgery, compared with their preoperative levels. At one month, five subjects showed a mean 0.79 ± 0.66 μg/mL increase in irisin levels and seven had a mean 0.26 ± 0.12 μg/mL decrease, compared with the preoperative levels. The preoperative irisin levels were not different between these two groups, although their pattern of alteration showed a significant difference in the two-way RM ANOVA analysis (*p* = 0.009), and the mean serum irisin levels at 1 month showed a significant difference using the Mann–Whitney *U* test (*p* = 0.007). Nine months after surgery, eight subjects who had lower preoperative irisin levels showed a statistically significant elevation in the mean irisin level of 0.18 ± 0.14 μg/mL, while the other four did not. We observed no significant difference in age, sex distribution, prevalence of diabetes mellitus, or hypertension medication history when comparing the two groups with either increases or decreases in irisin levels one or nine months after surgery. Preoperative irisin levels had a Spearman’s rank correlation coefficient (rho) of 0.524 (*p* = 0.08) with irisin levels at nine months after RYGBP. Using Friedman’s F test, there was no statistical difference among the three time points.

### 3.3. Changes in Irisin Levels and Weight Reduction

We classified the patients according to their elevation or reduction in irisin levels after RYGBP. First, we analysed the association of a change in irisin level during the initial month with weight reduction by subgroup analysis (Figure 2). The seven subjects of the group with decreased irisin levels one month post-surgery appeared to have higher %EWL three months post-surgery (*p* = 0.088) than the group with increased irisin levels; however, the longitudinal one-year changes in %EWL were not significantly different. The five subjects with increased irisin levels had a higher mean preoperative BMI than the group with decreased irisin levels (42.9 ± 2.4 vs. 38.9 ± 4.5 kg/m^2^; *p* = 0.042), although the change in BMI one year post-surgery was not significantly different between the two groups (*p* = 0.108).

Next, we compared weight reduction between the two groups with and without elevated irisin levels nine months after RYGBP (Figure 3). The mean preoperative irisin level in the first group (0.89 ± 0.09 μg/mL) was significantly lower than in the group with an unchanged level (1.26 ± 0.22 μg/mL; *p* = 0.016). The second group, with a higher preoperative irisin level (*n* = 4) had a significantly higher %EWL (*p* = 0.006) compared with the group with an elevated postoperative level but a lower preoperative irisin level (*n* = 8).

### 3.4. Association of Preoperative Irisin Levels with Weight Reduction after RYGBP

To analyse the association of preoperative serum irisin levels with weight reduction after RYGBP, we evaluated preoperative irisin levels and %EWL. As shown in Figure 4, preoperative irisin levels were correlated with %EWL at three months (β = 0.573; R^2^ = 0.478; *p* = 0.069, Figure 4A) and one year (β = 0.576; R^2^ = 0.612; *p* = 0.04, Figure 4B) after RYGBP, although the significance levels were marginal.

## 4. Discussion

In this pilot prospective observational study, we measured circulating irisin concentrations at three time points—preoperatively, at one month, and at nine months after RYGBP—and analysed their associations with weight reduction. We decided to measure the irisin concentrations on, before, and at one and nine months after surgery because we hypothesized that signals from muscles could contribute to the weight reduction effects of bariatric surgery and that surgery could have an influence on the signaling of muscles. We assumed that abrupt alteration of muscle metabolism and gut signaling induced by surgery could have an influence on myokines like irisin. Circulating irisin levels could reflect muscle metabolism or muscle mass, and hence, circulating irisin levels before and one and nine months after RYGBP could be an indicator of muscle function before, during, and after weight reduction by RYGBP. In the first three months after RYGBP, patients experienced very low energy intake and muscle wasting accompanying fast weight reduction; irisin levels at one month after surgery may represent this period. Between 6 and 12 months after RYGBP, most patients’ energy intake and body weights had stabilized, and irisin levels nine months after RYGBP may reflect this period.

Although the number of samples was limited, our data showed interesting findings. First, we found that serum irisin concentrations changed bidirectionally one month after RYGBP. Thus, seven patients with decreased irisin values one month post-surgery tended to have a higher %EWL three months post-surgery. Second, eight of the patients who showed lower preoperative irisin levels and elevated levels nine months after RYGBP had smaller weight reduction than the four patients who showed no postoperative increase. Third, by nine months, the bidirectional changes in irisin levels observed at one month post-surgery returned to a similar range as before surgery. Fourth, the preoperative irisin levels were significantly associated with %EWL by regression analysis.

The patients who underwent RYGBP experienced loss of appetite and indigestion for a few months and recovered slowly over time. One month after RYGBP is usually a period of severe negative nitrogen balance that might influence muscle physiology and metabolism [30]. Because of the very low energy intake and malabsorption in this early postoperative period, this negative nitrogen balance is accompanied by rapid losses in weight and LBM. Energy deprivation and protein deficiency result in decreased protein synthesis in skeletal muscles during this period [31,32]. Unexpectedly, the alteration of irisin one month post-RYGBP occurred bidirectionally in this group. Five of the subjects showed irisin values that increased by 88.9 ± 75.9%, ranging from 1.04 to 2.51 μg/mL, while seven of them had irisin levels that decreased by 23.0 ± 7.1%, in a range of 0.55–1.16 μg/mL. This dispersal in the results was reversed when the patients had recovered from energy depletion and negative nitrogen balance, and serum irisin levels nine months after surgery returned to similar ranges as seen preoperatively. A prolonged negative nitrogen balance is usually associated with lower caloric intake, and this could have caused the greater weight reductions in the four patients who did not show elevated irisin levels nine months post-surgery.

Previous studies [33,34,35,36,37,38,39] reported inconsistent results on the relationship between circulating irisin and body mass. One study [35] reported that circulating irisin concentrations in bariatric patients decreased along with LBM loss six months after surgery. However, another study [36] reported opposing results among morbidly obese subjects preparing for bariatric surgery who had lower serum irisin levels than non-obese subjects. In other studies, irisin levels were not proportional to BMI or LBM [24,37]. After losing weight and relieving metabolic burdens, patients subjected to bariatric surgery tended to have physically active lifestyles, and their muscular fitness improved with physical training, which could prevent the reduction in irisin levels even if they had lost LBM. Circulating irisin levels might reflect not only LBM and BMI but also lifestyle interventions, including adjustments to nutritional balance and increased muscle fitness [38]. A negative nitrogen balance is usually concomitant with a lower caloric intake in patients after bariatric surgery, and this could lead to greater weight reductions and decreased irisin levels. Therefore, detailed measurements of confounding factors affecting changes in serum irisin levels after bariatric surgery are needed to interpret these variations over time, particularly in the subacute postoperative period (e.g., one month postoperatively). Morbidly obese patients with similar BMI values have different degrees of muscle fitness. Therefore, we hypothesize that this could reflect the level of irisin, which in turn could influence the effects of surgery on weight loss [39,40,41]. If future studies could confirm that serum irisin levels reflect muscle fitness along with muscle mass, this assumption would be valid.

On the other hand, irisin has proven to be related to metabolic parameters and adiposity as not only a myokine but also as an adipokine in various studies. Circulating irisin levels increased nearly 10% after an average 27 kg of TFM loss (27.0 ± 8.3 kg, 33.6 ± 8.1% of total fat) following bariatric surgery in our study, while irisin levels in humans were positively correlated with parameters of adiposity [34,35] and associated with markers of glucose and lipid homeostasis disturbance in obesity and in patients with metabolic syndrome [39,42]. A recent study reported that human visceral and subcutaneous adipose tissues in obese patients secrete irisin, and the blocking of irisin gene expression is related to reduced UCP1 (uncoupling protein 1) expression and enhanced adipogenesis [43]. Therefore, we could not exclude the possibility that differential acute loss of adiposity in our patients caused the bidirectional alteration of irisin concentration one month after surgery.

This prospective observational pilot study had several limitations. The first is the small sample size, particularly in sub-group analysis. Of the 25 subjects in the eligible population, only 12 met the enrolment criteria. Since the number of subjects in the sub-groups to compare variables was limited, we did not conduct a power analysis. Therefore, our results could not conclude the predictive value of preoperative or postoperative irisin concentration in RYGBP-induced weight reduction, rather, it could provide basic clinical data for the RYGBP-mediated alteration of irisin and weight reduction. Second, the heterogeneity of this population should be considered. Obesity has multiple causes, and the clinical status of obesity is multidimensional. Patients with similar levels of adiposity can have variable metabolic impairments and obesity-related endocrine dysfunctions. A third limitation is the variation in the clinical intervention needed by each patient. Medications for inducing further weight loss or controlling metabolic diseases could influence clinical outcomes. Nonetheless, this study has significance as a pilot study to generate a hypothesis on the effect of irisin on the clinical weight reduction by RYGBP. Further large-scale studies are required to test this hypothesis.

## 5. Conclusions

Preoperative fasting serum irisin concentration in obese patients who underwent RYGBP, may be associated, at least in part, with weight reduction after RYGBP in this pilot study. Although the number of samples was limited, our results suggest that the serum level of irisin before and after RYGBP could be a predictive biomarker for the weight reduction effects of surgery, which warrants further clinical studies with a large number of samples.

## Figures and Tables

**Figure 1 ijerph-16-00660-f001:**
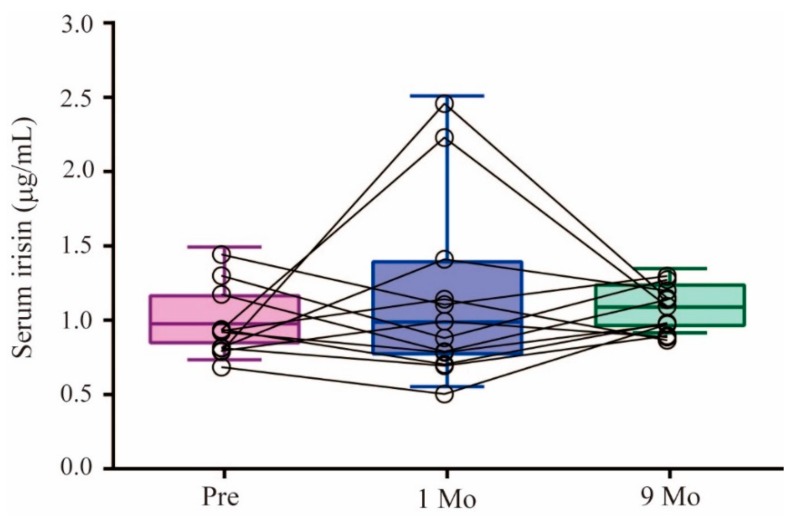
Changes in preoperative (Pre) circulating serum irisin concentrations with respect to the levels one (1 Mo) and nine months (9 Mo) after RYGBP; *p* = 0.0528 by Friedman’s F test.

**Figure 2 ijerph-16-00660-f002:**
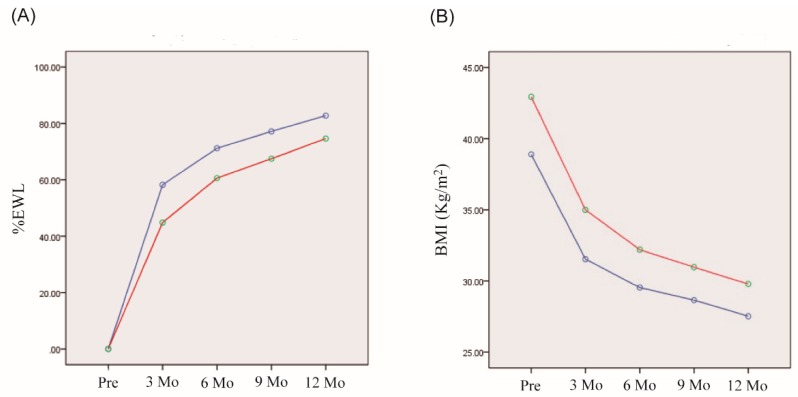
Comparison of the body-weight lowering effects of RYGBP during one year between two groups with increased or decreased irisin levels one month after RYGBP. (**A**) Changes in %EWL. (**B**) Changes in BMI. Blue and red lines represent groups with increased or decreased irisin levels one month after RYGBP, respectively. No significant differences were found between the groups (A, *p* = 0.296; B, *p* = 0.108 by 2-way RM ANOVA).

**Figure 3 ijerph-16-00660-f003:**
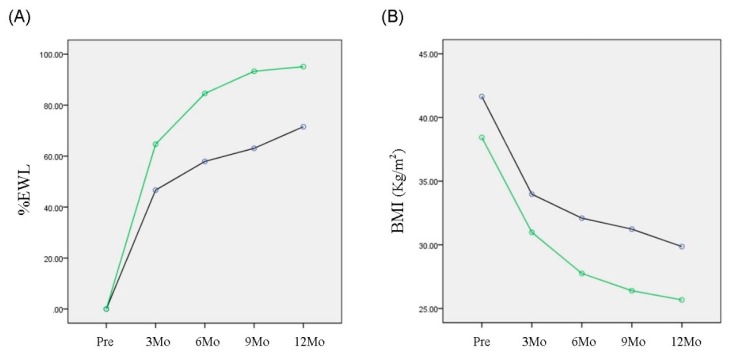
Comparison of the body-weight lowering effect of RYGBP one year after surgery between groups with or without elevated irisin levels detected nine months after RYGBP. Black and green lines represent groups with or without elevated irisin levels nine months after RYGBP, respectively. (**A**) The %EWL in the group without an elevated irisin concentration was larger than in the group with elevated irisin levels (*p* = 0.006 by two-way RM ANOVA). (**B**) The decrease of BMI in the group without an elevated irisin level was significantly greater than in the group with an elevated level (*p* = 0.032 by two-way RM ANOVA).

**Figure 4 ijerph-16-00660-f004:**
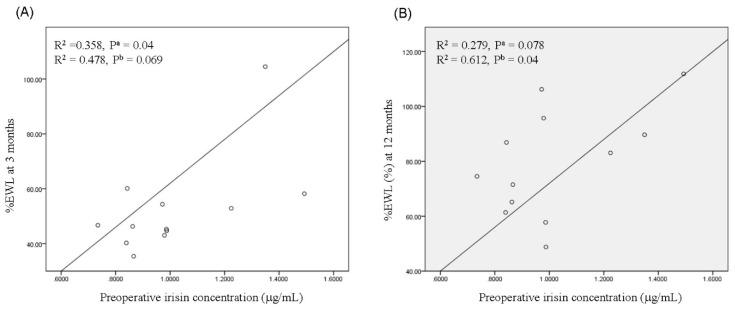
Correlation of preoperative serum irisin levels with %EWL three months (**A**) and one year (**B**) after RYGBP. The *p* values are indicated for linear regression analyses of data. ^a^ unadjusted or ^b^ adjusted for age and gender.

**Table 1 ijerph-16-00660-t001:** Longitudinal changes in clinical characteristics after Roux-en-Y gastric bypass surgery (RYGBP) (*n* = 12).

	Pre	3 Months	6 Months	9 Months	12 Months	*p*-Value ^a^
Weight (Kg)	114.4 ± 18.7	92.8 ± 14.1	86.3 ±12.6	83.5 ± 13.2	80.5 ±15.0	<0.001
BMI (kg/m^2^)	40.6 ± 4.2	32.9 ± 3.3	30.6 ± 3.0	29.6 ± 3.24	28.1 ± 3.49	<0.001
AC (cm)	119.0 ±14.3	106.3 ± 12.4	100.4 ± 10.9	97.2 ± 10.3	94.7 ± 11.6	<0.001
%EWL	NA	52.7 ± 17.9	66.8 ± 16.4	73.1 ± 19.3	79.4 ± 19.6	<0.001
SBP (mmHg)	136.4 ±21.6	121.4 ± 6.7	124.6 ± 9.6	119.3 ± 10.0	117.2 ± 11.1	0.006
DBP (mmHg)	87.2 ± 16.4	75.6 ± 5.7	77.3 ± 6.4	75.2 ± 8.0	74.4 ± 10.2	0.001
TG (mg/dL)	181.3 ± 135.6	108.7 ± 52.4	105.3 ± 53.0	85.6 ± 26.7	84.3 ± 25.7	<0.001
HDL (mg/dL)	42.6 ± 10.1	41.7 ± 9.2	50.6 ± 11.9	53.7 ± 13.3	55.6 ± 12.8	<0.001
Fasting glucose (mg/dL)	103.7 ± 20.7	94.2 ± 14.4	93.4 ± 10.6	92.8 ± 11.1	90.3 ± 11.1	0.055
HbA1C (%)	6.19 ± 1.04	5.45 ± 0.48	5.35 ± 0.41	5.39 ± 0.37	5.27 ± 0.38	<0.001
Fasting insulin (µU/mL)	18.2± 7.04	10.9 ± 3.8	10.7± 5.4	9.8± 2.8	10.1 ± 3.1	0.002

^a^ Friedman’s F test. Abbreviations: Pre; preoperative baseline level; NA, not available; BMI, body mass index; AC, abdominal circumference; %EWL, percentage of excess bodyweight loss; SBP, systolic blood pressure; DBP, diastolic blood pressure; TG, triglyceride; HDL, high-density lipoprotein; HbA1C, hemoglobin A1C.

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
