# Peer review of "Association of Circulating Irisin Concentrations with Weight Loss after Roux-en-Y Gastric Bypass Surgery"

_ijerph, 2019, doi:10.3390/ijerph16040660_

Round 1
Reviewer 1 Report
Comments:
The manuscript topic is interesting- to study the effect of Weight Loss
After Roux-en-Y Gastric Bypass Surgery on irisin concentrations, however the major concerns are about methodology.
According my opinion 12 study subjects is not enough to get appropriate statistical power, therefore I would recommend to do power analysis and put this facts to the article to section Methods. They observed 2 groups with either increases or decreases in irisin levels at 1 or 9 months after surgery, however 1 group comprise only 5 subjects, second only 7 subjects.
The major concern and limitation of the study is the number of samples. Therefore results and conclusions should be alleviated.
Author Response
Dear Reviewer and Editor,
First of all, all authors appreciate the valuable comments by Reviewer. We carefully reviewed the comments suggested by Reviewer 1 and respond to the comments, as below. The major issue is the limited sample size in our study, and we agree on this. Although our study could not reach the solid conclusion for the predictive value of pre-operative irisin concentration or change of irisin level by surgery to expect post-operative weight reduction, our results suggested the association of pre-operative irisin concentration with RYGBP and descriptively provide clinical data for the RYGBP-mediated alteration of irisin and weight reduction. We edited several sentences in Abstract and Conclusion, as well as sentences in Discussion section.
All modified sentences or paragraph below (red-colored text) will be found in the text of revised version of manuscript as well.
We believe that the revised version is significantly improved by following the Reviewer’s comments. Please review our responses below.
Best Regards,
Ju-Hee Kang, M.D., Ph.D.
Department of Pharmacology, College of Medicine, Inha University
Incheon, 22212, Republic of Korea

Reviewer 2 Report
This is an interesting study, aiming to assess associations of irisin levels with weight loss after gastric bypass surgery. Overall, the study is well-designed and good report.
However, it is desirable that the authors report more details to help readers evaluate the work and for planned follow-up studies seeking to replicate or build-on the reported associations. Generally the writing is clear.
The authors do not provide any explanation why they took to measure serum irisin concentrations before, and at 1 and 9 months after surgery. They observed bidirectional 2 groups with either increases or decreases in irisin concentrations at 1 or 9 months after surgery when they compared with their preoperative concentrations. There is no description of how they differently have demographic characteristics before, and at 1 and 9 months after surgery. Please, compare and report.
It is not clear why the authors investigated the associations of preoperative irisin levels with weight reduction at 3 and 12 months after surgery. At 1 or 9 months after surgery, serum irisin levels showed bidirectional pattern in Figure 1. Irisin levels at 9 month (Figure 3) after surgery did show the body-weight lowering effects of surgery through 1 year, but not at 1 month (Figure 2). Did the authors consider the changes of serum irisin levels between before and 1 month or 9 month and the effects of weight reduction after surgery?
The authors discussed generally about irisin regulation after bariatric surgery and muscle fitnesses along with muscle mass, although irisin secreted from subcutaneous, visceral, and epididymal adipose tissue and also known as adipokine. Please, discuss more on irisin studies in morbid obese, by discussing specific similarities/differences in experimental design/results with the present study.
Typo error: finesses in page 7.
Author Response
Dear Reviewer and Editor,
First of all, all authors appreciate the valuable comments by Reviewer. We carefully reviewed the comments suggested by Reviewer 2 and respond to the comments, as below.
We added several sentences to discuss the issues that Reviewer pointed out in Discussion section, and we believe that the revision will provide the clear understanding the results and values of our study. In addition, we added a paragraph discussing the source of irisin and the effects of adipose-derived irisin on our result interpretation.
All modified sentences or paragraph below (red-colored text) will be found in the text of revised version of manuscript as well.
We believe that the revised version is significantly improved by following the Reviewer’s comments. Please review our responses below.
Best Regards,
Ju-Hee Kang, M.D., Ph.D.
Department of Pharmacology, College of Medicine, Inha University
Incheon, 22212, Republic of Korea

Round 2
Reviewer 2 Report
This version is substantially improved. The authors need to correct all sentence errors (e.g. line 206; what's level?, line 251; acute loss of adiposity in your patients differntiated your patients? etc) according to the basic rules of grammar.Author Response
We appreciate the reviewer's comment, and edited the sentences to improve the clarity of sentences. The changes of words were marked as red-colored text in the revised manuscript.